# Task-Oriented Multi-Bitstream Optimization for Image Compression and Transmission via Optimal Transport

## ABSTRACT

Image compression for machine vision exhibits various rate-accuracy performance across different downstream tasks and content types. An efficient utilization of constrained network resource for achieving an optimal overall task performance has thus recently attracted a growing attention. In this paper, we propose *Tombo*, a task-oriented image compression and transmission framework that efficiently identifies the optimal encoding bitrate and routing scheme for multiple image bitstreams delivered simultaneously for different downstream tasks. Specifically, we study the characteristics of image rate-accuracy performance for different machine vision tasks, and formulate the task-oriented joint bitrate and routing optimization problem for multi-bitstreams as a multi-commodity network flow problem with the time-expanded network modeling. To ensure consistency between the encoding bitrate and routing optimization, we also propose an augmented network that incorporates the encoding bitrate variables into the routing variables. To improve computational efficiency, we further convert the original optimization problem to a multi-marginal optimal transport problem, and adopt a Sinkhorn iteration-based algorithm to quickly obtain the near-optimal solution. Finally, we adapt *Tombo* to efficiently deal with the dynamic network scenario where link capacities may fluctuate over time. Empirical evaluations on three typical machine vision tasks and four real-world network topologies demonstrate that *Tombo* achieves a comparable performance to the optimal one solved by the off-the-shelf solver Gurobi, with a 5× ∼ 114× speedup.

## CCS CONCEPTS

• **Information systems → Multimedia streaming**; • **Networks → Traffic engineering algorithms**.

## KEYWORDS

Task-oriented image compression and transmission, variable bitstream, multi-commodity flow problem, optimal transport.

## 1 INTRODUCTION

To relieve contradictions between the ever-growing Internet traffic and constrained bandwidth of real-world communication networks, compressing data before delivering them for downstream tasks becomes an efficient and popular solution, which alleviates network congestion and reduces transmission delay [6, 7]. Among the

Internet traffic globally, visual streams have accounted for over 65% of the total amount [27]. Thus, studies on the efficient compression and transmission for these visual data, such as images and videos, have recently attracted a lot of attention [5, 8, 24].

Traditional image-coding frameworks have evolved over decades, aiming to solve a rate-distortion optimization problem. In lossy image compression, this optimization can be achieved by a trade-off between the bitrate and distortion, where the distortion is usually measured by peak signal-to-noise ratio (PSNR) or multiscale structural similarity (MS-SSIM) [30]. However, as more and more image traffic is now used for machine vision tasks, such as classification and detection, rather than for human watching, how to achieve the optimal rate-accuracy performance for a specific machine vision task has become an emerging research direction. Along this, various end-to-end learnable compression systems have shown their superior performance for task-oriented image coding [6, 22]. For example, Cui *et al.* in [11] propose a rate-adjustable learned image compression framework, providing a continuously variable bitrate for image bitstream to adapt to time-varying network throughput.

Unfortunately, these existing studies on variable bitstream image compression assume that the *optimal encoding bitrate* is known in advance or determined by the estimated link capacity, without taking into account the subsequent network transmissions. When multiple image bitstreams are requested to be delivered simultaneously over the network, transmission cost (e.g., delay, jitter or loss) may inevitably increase, leading to a degradation on the quality of service for downstream tasks [4, 24]. Note that traffic engineering (TE) techniques can find the *optimal routing scheme* for these bitstreams to achieve a desired transmission performance with given traffic demands [1]. However, separately employing the bitrate allocation for image compression and TE solution for routing results in a poor performance, or even negative effect [26]. To make a full utilization of network resources for the need of specific downstream tasks, it is thus imperative to identify a compression-and-transmission optimization scheme that jointly determines the optimal bitrate allocation and routing scheme for requested image bitstreams. Recently, steps [14, 26, 34] have been taken in joint optimization of the encoding rate and routing scheme for adaptive video streaming, which, however, ignore differences in the rate-accuracy performance between different machine vision tasks.

In this paper, we propose *Tombo*, a task-oriented multi-bitstream optimization framework that determines the optimal image encoding bitrate and routing scheme for each transmission request, according to the downstream task requirements and network conditions. Specifically, we study the characteristics of image rate-accuracy performance for different machine vision tasks, with the widely used end-to-end image compression method. Then, we formulate the task-oriented multi-bitstream image compression and transmission problem as a multi-commodity flow (MCF) problem using a time-expanded network model. This problem optimizes

the image bitrates and corresponding transmission paths, aiming to strike a trade-off between minimizing the overall network link utilization and maximizing the overall downstream task performance within a pre-defined transmission delay. Though off-the-shelf solvers, e.g., Gurobi [15] and CPLEX [10], can accurately solve this problem to get the optimal solution, they cannot scale to the growing number of network nodes and commodities. To make the solution feasible in practice, we further convert the MCF problem to a multi-marginal optimal transport problem, and smooth its feasible region by introducing an entropy regularization term. An extended version of the Sinkhorn algorithm is then adopted to obtain a near-optimal solution with high computational efficiency. For the dynamic network scenario with fluctuating link capacities over time, we further enhance *Tombo* to rapidly adapt to the network changes. Finally, we conduct experiments on three typical machine vision tasks and four real-world network topologies of different node sizes, demonstrating effectiveness of the proposed method in terms of the overall task performance and computational efficiency. Our main contributions can be summarized as follows.

- We propose *Tombo*, a joint optimization framework that performs task-oriented rate adaptation and routing for the encoding and transmission of multiple image bitstreams.
- We formulate the multi-bitstream image compression and transmission optimization problem as a multi-marginal optimal transport problem, and develop a computationally efficient algorithm based on Sinkhorn iterations to obtain a near-optimal solution.
- For dynamic networks with fluctuating link capacities, we enhance *Tombo* to efficiently adapt to these changes by leveraging solutions obtained in the previous transmission period.
- We evaluate *Tombo* on four real-world networks with three machine vision tasks. Compared to the state-of-the-art commercial solver Gurobi [15], *Tombo* achieves a comparable overall performance and yields a 5× ∼ 114× speedup.

## 2 PROBLEM FORMULATION

### 2.1 Image Compression for Machine Vision

Rate-distortion optimization is the core of lossy image compression. Classical image compression framework is usually built upon optimizing the objective function: $\mathcal{L} = R + \lambda D$, where $R$ is the bitrate of the compressed image, $D$ measures the distortion of the reconstructed image, and $\lambda$ controls the trade-off between the rate and distortion. In an end-to-end learned compression framework for machine vision tasks, $D$ is defined as the task-oriented distortion between the reconstructed features and the ground truth, which is measured by specific quality metrics, such as the accuracy for classification tasks, or the mean average precision (mAP) for detection and instance segmentation [7, 29, 33]. For example, the rate-performance curves of lossy image compression in Fig. 1 indicate that the trade-off between the rate and accuracy/mAP varies across different downstream tasks, which also differs between various image categories.

Fig. 2(a) shows a common scenario where a sequence of compressed image signals are requested to be transmitted from a source (e.g., a monitor) to several targets to fulfill different tasks (e.g., classification or object detection). The complex network structure in the

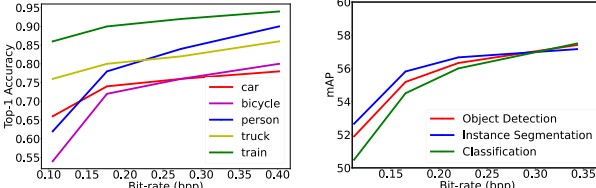

**Figure 1: Rate-performance curves presented in [7]: (left) rate-accuracy for various image categories, and (right) rate-mAP/accuracy for different vision tasks of Category "Bird".**

real world (e.g., existence of relays) offers multiple feasible paths for delivering these image bitstreams. In pursuit of the optimal task performance at targets, it is desirable to transmit as many bits as possible, since higher bitrates often yield better performance. However, simultaneously transmitting these image bitstreams over bandwidth-constrained networks may lead to network congestion, resulting in detrimental effects such as package loss, retransmission, and significant transmission delays that adversely impact the downstream tasks. On the other hand, traditional routing methods strive to optimize traffic allocation for given volumes of image bitstreams across available paths, to minimize the transmission cost caused by the traffic overload [26]. Thus, it is essential to strike a trade-off between the transmission cost and overall task performance for multi-bitstream compression-and-transmission scenarios.

### 2.2 Multi-Bitstream Compression and Transmission

We focus on the task-oriented optimization of encoding bitrates and routing paths for multiple image bitstreams associated with different downstream tasks and various image categories. Specifically, we consider a communication network with constrained link capacities, represented as a directed graph $\mathcal{G} = (\mathcal{V}, \mathcal{E})$, where nodes $v_i, i = 1, \cdots, |\mathcal{V}|$ represent the network sites and directed edges $e_{ij} = (v_i, v_j), i \neq j$ denote the links between these sites. Note that we will omit the subscripts of $e$ for notational simplicity throughout this paper, i.e., $e_{ij}$ written as $e$ in places where there is no ambiguity.

For task-oriented image transmission, a source wants to send one or multiple images to a target to perform a machine vision task, which is referred to as *traffic demand* of this source-target pair. Here, we assume that all traffic demands transmitted via the network must be completed *within a given duration of time* $\Gamma$, i.e., the maximum transmission delay. Thus, the optimization variables consist of both the amount of bits that needs to be sent and the routing paths to complete these traffic demands. Subsequent transmission of new traffic demands can initiate only upon completion of previous ones.

As shown in Fig. 2(a), we consider a common scenario with some network sites serving as source nodes, which send different categories of images to some target nodes that receive these image bitstreams for implementing various machine vision tasks. We represent each image bitstream as a commodity with index $l = \{1, \cdots, L\}$, and denote the task index by $\theta = \{1, \cdots, \Theta\}$. Under this setting, the joint bitrate allocation and routing optimization for multi-bitstream compression-and-transmission can be modeled as a multi-commodity flow (MCF) problem, where the objective is to find an appropriate encoding/transmission bitrate and routing strategy for each commodity to strike a balance between the network transmission cost and task performance with a given maximum transmission delay $\Gamma$. We assume that the transmission begins at

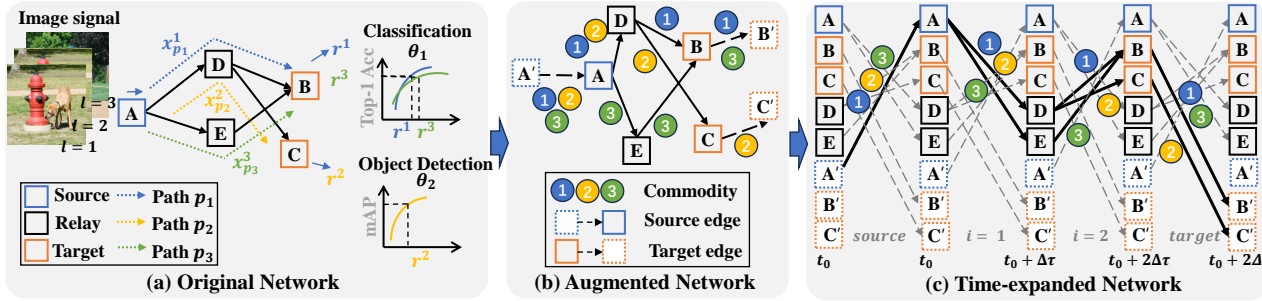

**Figure 2: Example network with 1 source, 2 relays and 2 targets, its augmented network and time-expanded network with $\Gamma = 2\Delta\tau$, where the source A sends two commodities to target B and one commodity to target C.**

time $t_0$. Specifically, we adopt the average overall network link utilization within the transmission period $[t_0, t_0 + \Gamma)$ as the metric of transmission cost, as widely used in the TE studies [26, 32]. Similar to the setting in [26], we use a linear-form function to calculate the instantaneous link utilization of link $e \in \mathcal{E}$ produced by all the commodities at time $t \in [t_0, t_0 + \Gamma)$, which can be expressed as: $\Psi_e(\vec{x}_{e,t}) = \sum_{l=1}^{L} x_{e,t}^l / d_{e_t}$, where $\vec{x}_{e,t} = \{x_{e,t}^1, ..., x_{e,t}^L\}$, $x_{e,t}^l$ denotes the transmission bitrate of the $l$-th commodity on link $e$ at time $t$, and $d_{e_t}$ represents the instantaneous bandwidth of link $e$ at time $t$. In addition, we represent the task performance of machine vision task $\theta$ w.r.t. the $l$-th commodity (i.e., image bitstream) by a convex function: $\Phi_{l,\theta}(r^l) = -a_{l,\theta}/(r^l - b_{l,\theta}) + c_{l,\theta}$, where $r^l$ is the received average bitrate of the $l$-th commodity at the target node within the transmission period $[t_0, t_0 + \Gamma)$. The value of this task performance function increases with the increase of $r^l$, and the parameters $a_{l,\theta}, b_{l,\theta}, c_{l,\theta}$ can be fitted based on any existing task-oriented image compression method. Therefore, the objective function of our multi-bitstream image compression and transmission optimization problem within the transmission period $[t_0, t_0+\Gamma)$ can be mathematically expressed as:

$$\min_{\vec{x}_{e,t}, r^l} \frac{1}{\Gamma} \int_{t_0}^{t_0+\Gamma} \sum_{e \in \mathcal{E}} \Psi_e(\vec{x}_{e,t}) \mathrm{d}t - \lambda' \sum_{l=1}^{L} \sum_{\theta=1}^{\Theta} \Phi_{l,\theta}(r^l), \quad (1)$$

where $\lambda'$ is a Lagrange multiplier that controls the trade-off between the transmission cost and task performance.

## 2.3 Bitrate Allocation via Routing Optimization

Notably, is intractable to optimize the variables in Eq. (1) with a continuous time index. Inspired by [2, 16, 21, 31], we thus introduce a discrete time-expanded network to reformulate the objective function and address the joint optimization problem in this paper. As illustrated in Fig. 2(c), the time-expanded network replicates the original topology at each discretized time interval, which serves as a singular acyclic directed graph integrating both the network topology and temporal aspects seamlessly [19, 28]. A feasible path in the original network can be depicted as a series of directed edges within the time-expanded network. In this context, the total transmission period $\Gamma$ is divided discretely into $\tau \in \mathbb{N}_+$ time intervals, with the duration of each interval expressed as a constant value $\Delta\tau = \Gamma/\tau \in \mathbb{R}_+$. Note that a smaller value of $\Delta\tau$ indicates a fine-grained time division in constructing the time-expanded network, which leads to a more precise optimization for the objective in Eq. (1), but also at the cost of a higher computational overhead.

Since optimization variables of the time-expanded network are the allocated traffic volumes for the links, it is challenging to simultaneously optimize the volumes of traffic demands (e.g., the encoding bitrate of image bitstreams), which are only reflected on the source and target nodes. To map the traffic volumes of the source and target nodes into the traffic volumes of the edges, we augment the number of edges from $|\mathcal{E}|$ to $n = |\mathcal{V}'| + |\mathcal{E}|$ by additionally adding specialized source nodes and target nodes to the original network. Here, $|\mathcal{V}'|$ is the total number of source and target nodes. The resulting new network topology is referred to as the augmented network. For example, as shown in Fig. 2(b), the received traffic volume $r^2$ by target node $C$ in the original network is equivalent to the traffic volume on the target edge $CC'$ in the augmented network.

We denote the set of all available paths of the $l$-th commodity over the time-expanded network as $P^l$, and a routing path of the $l$-th commodity for any source-target pair can be represented as $p = (e_+^p, e_1^p, \cdots, e_\tau^p, e_-^p) \in P^l$, i.e., a sequence of directed edges that originate from a source edge $e_+^p = e^+ \in \mathcal{E}^{l,+}$ and terminate at a target edge $e_-^p = e^- \in \mathcal{E}^{l,-}$, where $\mathcal{E}^{l,+}$ and $\mathcal{E}^{l,-}$ denote the sets of source edges and target edges, respectively. Moreover, we use $p[i] = e_i^p, i = 1, \cdots, \tau$ to represent the transport link of path $p$ at the $i$-th time interval. Then, we denote by $x_p^l \in \mathbb{R}_+$ the traffic volume allocated on path $p$ for the $l$-th commodity.

In Fig. 2(c), we highlight the routing paths from the source node $A$ to the target nodes $B$ and $C$ with bold arrows for the sake of clarity. The sent and received traffic volumes of all the three commodities within the period $[t_0, t_0+\Gamma)$ can be replaced with the traffic volumes on the source edges $e^+$ and target edges $e^-$, respectively. Thus, the variable $r^l$ in Eq. (1) can be explicitly determined by the variable $x_p^l$. The multi-bitstream image compression and transmission optimization problem with the time-expanded network modeling can thus be formulated as:

**P1:**
$$\min_{x_p^l} \sum_l \left( \sum_{p \in P^l} \Psi(x_p^l) - \lambda' \sum_{e^- \in \mathcal{E}^{l,-}} \sum_{\theta \in \mathcal{E}_{e^-}^{\mathrm{task}}} \Phi_{l,\theta}(r_{e^-}^l) \right) \quad (2a)$$

$$\text{s.t.} \quad \sum_{p \in P^l, \, p[0]=e^+} x_p^l = r_{e^+}^l, \, \forall\, e^+ \in \mathcal{E}^{l,+}, \, l = 1, \cdots, L, \quad (2b)$$

$$\sum_{p \in P^l, \, p[\tau+1]=e^-} x_p^l = r_{e^-}^l, \, \forall\, e^- \in \mathcal{E}^{l,-}, \, l = 1, \cdots, L, \quad (2c)$$

$$\sum_l \sum_{p \in P^l, \, p[i]=e} x_p^l \le \bar{d}_e, \, \forall e \in \mathcal{E}, i = 1, \cdots, \tau, \quad (2d)$$

where the function $\Psi(x_p^l) = \frac{1}{\tau} \sum_{i=1}^{\tau} x_p^l / \bar{d}_{p[i]}$ denotes the average link utilization over path $p$ within the transmission period $[t_0, t_0 + \Gamma]$ and $\bar{d}_{p[i]}$ is the average transmission capacity of link $p[i]$ within the time interval $t_0$ to $t_0 + i\Delta\tau$. In addition, $\mathcal{E}_{e^-}^{\text{task}} = \{\theta_{e^-}\}$ represents the set of tasks that will be conducted in the target edge $e^-$. Constraints in Eqs. (2b) and (2c) state the relationship between the routing paths $x_p^l$ and the traffic demands distributed at the source and target edges, where $r_{e^+}^l$ is the allowed maximum traffic volume originating from the source edge $e^+$ for the $l$-th commodity, and $r_{e^-}^l$ is the total received volume of bitstream at the target edge $e^-$ for the $l$-th commodity. Note that the optimized encoding bitrate of the $l$-th commodity is determined by the value of $r_{e^-}^l$, which can be alternatively derived from the values of $x_p^l$. Constraint Eq. (2d) guarantees that the total traffic volume of all image bitstreams on an edge $e$ does not exceed its capacity $\bar{d}_e$ at each time interval.

The optimization problem **P1** can be accurately solved using off-the-shelf solvers, such as Gurobi [15] and CPLEX [10]. However, as the values of $L$, $\Theta$, and $|\mathcal{E}|$ increase, the number of optimization variables in **P1** grows exponentially [17], resulting in a huge overhead in identifying the optimal solution. When new images are required to be compressed and transmitted, or in the practical scenarios where the network condition varies significantly, it is necessary to re-solve problem **P1**. Therefore, it is crucial to find a computationally efficient approach to solving this problem.

## 3 METHODOLOGY

To address the scability issue, in this section, we propose a multi-marginal optimal transport-based method to efficiently find a near-optimal solution to the problem **P1**. Specifically, we formulate problem **P1** as a discrete form of the multi-marginal optimal transport problem, and further introduce an entropy regularization term into the objective function to make the feasible region more smooth. Then, we adopt a variant of the Sinkhorn algorithm to quickly solve the multi-marginal optimal transport problem with only a limited number of iterations. Finally, we extend our method to the dynamic network scenarios where the link capacities fluctuate over time. The computation efficiency of our proposed *Tombo* algorithm is further enhanced by leveraging the correlation of solutions between two consecutive transmission periods, i.e., $[t_0, t_0 + \Gamma]$ and $[t_0 + \Gamma, t_0 + 2\Gamma]$.

### 3.1 Multi-Marginal Optimal Transport Problem

Optimal transport is a mathematical framework that focuses on finding the optimal transport plan to transfer mass from an initial distribution to a final distribution while considering the associated costs and constraints [25]. To represent the transport paths for the various image bitstreams, we introduce a tensor $M \in \mathbb{R}_+^{L \times n^{(\tau+2)}}$, which denotes the transport plan over $\tau$ time intervals in the time-expanded network as shown in Fig. 2(c). As mentioned in Section 2.3, $n = |\mathcal{V}'| + |\mathcal{E}|$ represents the total number of edges in the time-expanded network. A transport plan element $M_{l,\bar{p}}$ then denotes the traffic volume of the $l$-th image (i.e., commodity) on the transport path $\bar{p} = (\bar{e}_0^p, \bar{e}_1^p, ..., \bar{e}_\tau^p, \bar{e}_{\tau+1}^p)$, where $\bar{e}_i^p$ is the transport link of path $\bar{p}$ at the $i$-th time interval. $\bar{e}_0^p$ and $\bar{e}_{\tau+1}^p$ are source link and target link respectively as mentioned in Section 2.3. Note that $\bar{e}_i$ represents a transport link at time interval $i$ that starts from a certain node

and can reach any other nodes in the time-expanded network. This path is chosen without any specific constraints or limitations.

Thus, the elements of $M$ encompass all possible combinations of directed links within the network, though it is obviously not in accordance with the actual network topology. In order to eliminate infeasible solutions, we encode the topology in the objective function by assigning an infinitely high cost to the infeasible paths, rather than explicitly defining the set of feasible paths in constraints. As a result, we denote by $C \in \mathbb{R}_+^{L \times n^{(\tau+2)}}$ the cost tensor of the transport plan, with its element $C_{l,\bar{p}}$ defined as:

$$C_{l,\bar{p}} = \sum_{i=1}^{\tau} C_{l,\bar{e}_i^p}^u + \sum_{i=0}^{\tau} C_{\bar{e}_i^p, \bar{e}_{i+1}^p}^*, \quad C_{l,\bar{e}_i^p}^u = 1/\bar{d}_{\bar{e}_i} \quad (3)$$

where $C_{l,\bar{e}_i^p}^u$ denotes the transmission cost of $l$-th commodity on link $\bar{e}_i$ for transport path $p$ at time interval $i$, and $C_{\bar{e}_i^p, \bar{e}_{i+1}^p}^*$ is determined by the original network topology, i.e., $C_{\bar{e}_i, \bar{e}_{i+1}}^* = 0$ if link $\bar{e}_i$ is connected to link $\bar{e}_{i+1}$ in the original network and equals $\infty$ otherwise. Here, $\bar{d}_{\bar{e}_i}$ denotes the average capacity of link $\bar{e}_i$, which is fixed within the $i$-th time interval.

Then, the total transmission cost of transport plan $M$ is calculated as the inner product of tensor $C$ and $M$, i.e., $<C, M> = \sum_{l=1}^{L} \sum_{\bar{p}} C_{l,\bar{p}} M_{l,\bar{p}}$. Keeping the task performance term in the objective the same as in problem **P1**, problem **P1** is then reformulated as a multi-marginal optimal transport problem:

**P2:** $\min_M <C, M> -\lambda' \sum_l \sum_{\bar{e}_{\tau+1}} \sum_\theta \Phi_{l,\theta}([P_{-1,\tau+1}(M)]_{l,\bar{e}_{\tau+1}})$ (4a)

$$\text{s.t.} \quad P_{-1,0}(M) \leq R^{(-1,0)}, \quad (4b)$$

$$P_i(M) \leq \bar{d}, \quad \forall i = 1, ..., \tau, \quad (4c)$$

where the initial and final distributions are imposed on the joint projections of tensor $M$ on the corresponding two marginals $P_{-1,0}(M)$, $P_{-1,\tau+1}(M) \in \mathbb{R}^{L \times n}$. The two matrices depict the traffic distribution of each image bitstream on the augmented network at the initiation and conclusion of transmission. We denote by $[P_{-1,\beta}(M)]_{l,\bar{e}_\beta}$ the value of the $(l, \text{idx}_\beta)$-th element of the matrix $P_{-1,\beta}(M)$, where $\beta = 1, \cdots, \tau+1$, and $\text{idx}_\beta$ denotes index of the link $\bar{e}_\beta$ in the augmented network. The elements of $P_{-1,0}(M)$ and $P_{-1,\tau+1}(M)$ are calculated as:

$$[P_{-1,0}(M)]_{l,\bar{e}_0} = \sum_{\bar{p}, \bar{e}_1^p = \bar{e}_1} M_{l,\bar{p}}, \quad [P_{-1,\tau+1}(M)]_{l,\bar{e}_{\tau+1}} = \sum_{\bar{p}, \bar{e}_{\tau+1}^p = \bar{e}_{\tau+1}} M_{l,\bar{p}}.$$

And the link-capacity constraint in Eq. (4c) is imposed on the projection on the $i$-th marginal $P_i(M) \in \mathbb{R}^n$. Similarly, the values of elements of $P_i(M)$ are calculated as:

$$[P_i(M)]_{\bar{e}_i} = \sum_l \sum_{\bar{p}, \bar{e}_i^p = \bar{e}_i} M_{l,\bar{p}},$$

which denotes the sum of traffic on link $\bar{e}_i$ of the augmented network at time interval $i$. The vector $\bar{d} \in \mathbb{R}^n$ in Eq. (4c) represents the link capacities, and $R^{(-1,0)} \in \mathbb{R}^{L \times n}$ in Eq. (4b) denotes the maximum allowable traffic volume for each source node to transmit, which is associated with $r_{e^+}^l$ in Eq. (2b). For the sake of brevity, we use $G(\cdot)$ to denote the function $\lambda' \sum_{l=1}^{L} \sum_{\bar{e}_{\tau+1}} \sum_\theta \Phi_{l,\theta}(\cdot)$ in Eq. (4a) in the following sections.

## 3.2 Near-Optimal Solution with Sinkhorn

An effective approach to solve the multi-marginal optimal transport problem is to smooth the original problem by adding an entropy regularization term [3]: $D(M) = \sum_{l,\bar{p}}(M_{l,\bar{p}}\log M_{l,\bar{p}} - M_{l,\bar{p}} + 1)$. Introducing such an entropy term modifies the feasible region boundaries of the optimization problem, resulting in smoother boundaries and facilitating the identification of the optimal solution. As its regularization parameter $\epsilon$ approaches zero, the obtained solution converges towards the ground truth [25]. Consequently, we add the entropy-regularized term to the objective of problem **P2**, and the corresponding optimization problem is expressed as:

$$\textbf{P3:} \quad \min_{M} <C, M> +\epsilon D(M) - G(P_{-1,\tau+1}(M)) \tag{5a}$$

$$\text{s.t.} \quad \text{Eqs. (4b) and (4c)}, \tag{5b}$$

where $\epsilon > 0$ is the regularization parameter, making the objective function $\epsilon$-strongly convex. This problem can be solved via a variant of the computationally efficient method, i.e., Sinkhorn scheme [12].

Concretely, the optimal solution to problem **P3** has been proved to take the form

$$M = K \odot U, \quad K = \exp(-C/\epsilon) \in \mathbb{R}^{L \times n^{(\tau+2)}}, \tag{6}$$

where $\odot$ denotes the Hadamard product, and the elements of $K$ are calculated as $K_{l,\bar{p}} = \exp(-C_{l,\bar{p}}/\epsilon)$. Similar to the decomposition in Eq. (3), $K_{l,\bar{p}}$ is decomposed as:

$$K_{l,\bar{p}} = \left(\prod_{i=1}^{\tau}[K^u]_{l,\bar{e}_i^p}\right)\left(\prod_{i=0}^{\tau}[K^*]_{\bar{e}_i^p \bar{e}_{i+1}^p}\right),$$
$$K^u = \exp(-C^u/\epsilon), \quad K^* = \exp(-C^*/\epsilon). \tag{7}$$

In addition, $U \in \mathbb{R}^{L \times n^{(\tau+2)}}$ can be decomposed as:

$$[U]_{l,\bar{p}} = [U^{(-1,0)}]_{l,\bar{e}_0}[U^{(-1,\tau+1)}]_{l,\bar{e}_{\tau+1}}\prod_{i=1}^{\tau}[u_i]_{\bar{e}_i}, \tag{8}$$

where $U^{(-1,0)} \in \mathbb{R}^{L \times n}$ and $u_i \in \mathbb{R}^n$ can be iteratively updated. Here, we introduce two iterative variables $\psi_i$ and $\phi_i$ to simplify the calculation of the projections of tensor $M$, i.e., $P_i(M), P_{-1,0}(M)$ and $P_{-1,\tau+1}(M)$. Then, the projections of tensor $M$ are given by:

$$P_i(M) = u_i \odot (\psi_i \odot \phi_i \odot K^u)^T 1,$$
$$P_{-1,0}(M) = U^{(-1,0)} \odot \psi_0, \quad P_{-1,\tau+1}(M) = U^{(-1,\tau+1)} \odot \phi_{\tau+1}, \tag{9}$$

where $U^{(-1,0)}$ and $u_i \in \mathbb{R}^n$ can then be iteratively computed via:

$$U^{(-1,0)} \leftarrow \min(R^{(-1,0)}./\psi_0, 1), \tag{10a}$$

$$u_i \leftarrow \min(\bar{d}./((\psi_i \odot \phi_i \odot K^u)^T 1), 1), \tag{10b}$$

where operator ./ denotes the element-wise division. The other component $U^{(-1,\tau+1)} \in \mathbb{R}^{L \times n}$ is updated via solving the equation:

$$0 = -U^{(-1,\tau+1)} \odot \phi_{\tau+1} + \partial(G)^*(-\epsilon\log(U^{(-1,\tau+1)})), \tag{11}$$

where $(G)^*$ denotes the Fenchel conjugate of function $G$.

Consequently, our proposed Sinkhorn iteration-based algorithm for quickly solving problem **P3** is summarized in Algorithm 1. Without loss of generality, we define a tolerance threshold $\eta$ for the iterative variable $U^{(-1,\tau+1)}$, in order to determine when to terminate the iterations illustrated in Algorithm 1, Lines 4–18. Let $\Delta U^{(-1,\tau+1)}$ be the difference in $U^{(-1,\tau+1)}$ between the values of two consecutive update iterations. If $\Delta U^{(-1,\tau+1)} \leq \eta$, the iterations in Algorithm 1 are terminated. While implementing our method, the value of $\eta$

---

**Algorithm 1** *Tombo*: the Sinkhorn iteration of solving problem **P3**

1: Initialize $u_1, ..., u_\tau, U^{(-1,0)}, U^{(-1,\tau+1)}$.     ▷ *Initialization*
2: $\psi_\tau \leftarrow U^{(-1,\tau+1)}K^{*T}$
3: **for** $i = \tau - 1$ to $0$ **do**
4:     $\psi_i \leftarrow (\psi_{i+1} \odot K^u)\text{diag}(u_{i+1})K^{*T}$ ▷ *Obtain initial values of $\psi_i$*
5: **end for**
6: **while** $\Delta U^{(-1,\tau+1)} \geq \eta$ **do**
7:     Update $U^{(-1,0)}$ by Eq. (10a)
8:     $\phi_1 \leftarrow U^{(-1,0)}K^*$
9:     **for** $i = 1$ to $\tau$ **do**
10:       Update $u_i$ by Eq. (10b)
11:       $\psi_{i+1} \leftarrow (\psi_i \odot K^u)\text{diag}(u_i)K^*$   ▷ *Recompute $\psi_i$ when $u_i$ is updated*
12:     **end for**
13:     Update $U^{(-1,\tau+1)}$ by Eq. (11)
14:     $\phi_\tau \leftarrow U^{(-1,\tau+1)}K^{*T}$
15:     **for** $i = \tau$ to $1$ **do**
16:       $\phi_{i-1} \leftarrow (\phi_i \odot K^u)\text{diag}(u_i)K^{*T}$
17:     **end for**
18: **end while**
19: **return** $u_1, ..., u_\tau, U^{(-1,0)}, U^{(-1,\tau+1)}$
20: /* *The solution to **P3** can then be derived based on Eqs. (6)–(8).* */

---

can be customized to achieve varying trade-offs between the convergence rate and computational efficiency.

## 3.3 Extension to Dynamic Network Scenarios

When there is a change in network capacity, it is common to collect samples of the new link capacities and then re-solve the optimization problem from scratch for the new network scenario. To enhance the computational efficiency of our *Tombo* in these dynamic network scenarios, we provide an extended version **Tombo-D**, which is capable of handling fluctuations in network capacities.

Specifically, we assume that the link capacities change dynamically between a transmission period $\rho_1 = [t_0, t_0 + \Gamma]$ and its subsequent transmission period $\rho_2 = [t_0 + \Gamma, t_0 + 2\Gamma]$. As mentioned in Eqs. (3) and (7), the matrix $K^u$ in solving problem **P3** is associated with the link capacities. We denote by $K^u_{\rho_1}$ the value of $K^u$ within the period $\rho_1$, and $K^u_{\rho_2} = K^u_{\rho_1} + \Delta K^u$ the value of $K^u$ within the period $\rho_2$. Meanwhile, it is shown in Algorithm 1 that the initial values of $\psi_i$ is of significance to the convergence rate of variables $u_i, U^{(-1,0)}$ and $U^{(-1,\tau+1)}$. Therefore, it is intuitive to leverage the prior knowledge of $K^u_{\rho_1}$ and the associated variable $\psi_i^{\rho_1}$ to speed up the iteration process of $\psi_i^{\rho_2}$, and then the update of $U^{(-1,0)}$ and $U^{(-1,\tau+1)}$, rather than attempting to update their values starting from a random initial point. Referring to the details given in Appendix A, we can replace the initialization of $\psi_i^{\rho_2}$ in Algorithm 1, Line 4 with the following equation:

$$\psi_i^{\rho_2} = \psi_i^{\rho_1} + (\psi_{i+1}^{\rho_1} \odot \Delta K^u)\text{diag}(u_{i+1})K^{*T}. \tag{12}$$

Here, the values of $\psi_i^{\rho_1}$ are re-used to establish good initial values that are close to the optimal values for the updates of $\psi_i^{\rho_2}$. The improved initialization of variable $\psi_i$ results in a reduced number of iterations for the process illustrated in Algorithm 1, Lines 4–18.

# 4 PERFORMANCE EVALUATION

## 4.1 Experiment Setup

We implement our proposed *Tombo* and other comparison baselines, and evaluate their performance in terms of the overall task performance and computational efficiency with three typical machine vision tasks. All the experiments are conducted on a desktop equipped with a 12-core Intel Xeon E5-2620 Processor and 32GB DDR4 DRAM. The experiment settings are outlined as follows.

*4.1.1 Network Topologies.* We conduct the experiments over four real-world network topologies, each with varying link capacities. The number of nodes, directed edges, selected source nodes and target nodes for these test networks are listed in Table 1, while the corresponding network topologies are shown in Fig. 3(a).

*4.1.2 Rate-Performance Function.* For different categories of images, we employ the end-to-end learning task-oriented image compression framework in [7] to generate the rate-accuracy curves for the classification task, and rate-mAP curves for object detection and instance segmentation tasks. The test image dataset used in our experiments is COCO2017 [23], where the average image size is $640 \times 480$ pixels. We fit the parameters of the function $\Phi_{l,\theta}(r^l)$ by using the task performance results in different discrete bitrate values. Examples are illustrated in Fig. 3(b). The average coefficient determination for parameter fittings is $R^2 = 0.97$, which verifies that the obtained parameters can well fit the actual data.

*4.1.3 Baselines.* We compare our method *Tombo* with the following four baselines: *1) **Tombo-Gurobi**:* a method that solves problem **P2** using the commercial solver Gurobi-v10.0.3 [15], the threads of parallel computation of Gurobi is set to 10; *2) **MCF-Gurobi** [13]:* a method that solves the original MCF problem in problem **P1**. Before starting the optimization, we use the Floyd-Warshall algorithm [9] to find the top-$k$ shortest paths for each source-target pair, and only optimize the traffic on these paths to improve computational efficiency; *3) **TM-Sinkhorn**:* a variant of *Tombo*, which fixes the transmitted bitrate of each image as the same value as the average bitrate of *Tombo*'s solution, and then optimizes the routing strategy using the Sinkhorn scheme; *4) **SRD-Sinkhorn**:* a variant of *Tombo*, which replaces the task-specific performance function $\Phi_{l,\theta}(r^l)$ with a task performance function that depends only on the bitrate value.

*4.1.4 Evaluation Metrics.* We first compare the comparison baselines in terms of three metrics, including 1) the total objective value: **Total Obj.** in Eq. (2a); 2) the transmission cost, which is calculated as **Trans. cost** $= \sum_{l=1}^{L} \sum_p \Psi(x_p^l)$; and 3) the surrogate task value: **Task-surr.** $= -\lambda' \sum_{l=1}^{L} \sum_{\theta=1}^{\Theta} [\Phi_{l,\theta}(r^l) - c_{l,\theta}]$. Specifically, the transmission cost indicates the transmission performance of the optimized routing strategy, while the surrogate task value measures the overall task performance across various images and downstream tasks. In addition, we measure the feasibility of the solutions generated by *Tombo*, through measuring the constraint violation on each link, i.e., **violation**$= \sum_i \| \max(0, P_i(M) - \bar{d}) \|_1$. Thus, a higher violation value indicates a higher probability that the obtained solution to the optimization problem is infeasible. For the computational efficiency, we estimate the wallclock time it takes for each comparison method to meet the termination criteria as the value of **running time**.

**Table 1: Details of the test network topologies.**

| Test Network | Node | Edge | Source | Target |
|---|---|---|---|---|
| B4 [18] | 12 | 38 | 3 | 3 |
| Janetbackbone [1] | 29 | 90 | 5 | 9 |
| Carnet [1] | 44 | 86 | 9 | 9 |
| UsCarrier [20] | 158 | 378 | 9 | 9 |

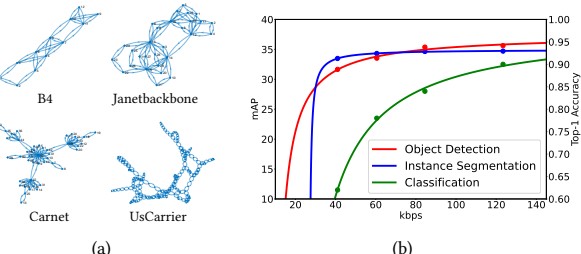

(a)  (b)

**Figure 3: Illustration of a) test network topologies, and b) rate-performance results and corresponding fitted curves.**

*4.1.5 Varying Link Capacity.* In order to simulate a dynamic network with fluctuating link capacities in experiments, we employ a random selection process to choose a proportion $\rho_{vl}$ of all network links. We then modify the capacities of these selected links from their original values $\bar{d}$ to random values within the range of $[(100-q)\%\bar{d}, (100+q)\%\bar{d}]$. In the four test network topologies, we keep the settings constant, i.e., $\rho_{vl} = 5\%$, $q = 2.5$, and conduct 10 repeatedly evaluations for the performance of *Tomob-D*.

*4.1.6 Other Parameters.* We set the Lagrange multiplier of the objective function as $\lambda' = 0.2$, and the number of images that are simultaneously compressed and transmitted as $L = \{9, 15, 27, 27\}$ for network B4, Janetbackbone, Carnet, and UsCarrier, respectively. We empirically set the total time interval $\Gamma$ as $\{7, 7, 8, 37\}$ milliseconds for four test networks, respectively and let $\Delta\tau$ always be 1 millisecond. As for the proposed *Tombo*, we set the entropy-regularized weight in problem **P2** as $\epsilon = 0.001$. The termination criteria for Gurobi is set as $\{1\%, 1\%, 1\%, 10\%\}$ primal-dual gap for network B4, Janetbackbone, and Carnet, and UsCarrier, respectively. We set the tolerance threshold of $U^{(-1,\tau+1)}$ as $\eta = 0.5$, and also set the maximum iteration number of the Sinkhorn iteration as $\{500, 1000, 1000, 1500\}$ for network B4, Janetbackbone, Carnet and UsCarrier, respectively. In addition, we further investigate the impact of different $\Delta\tau$ on the performance of *Tombo* in Appendix B.

## 4.2 Results

We compare *Tombo* with other baselines in terms of the objective value and running time performances in various network topologies and downstream task settings. Table 2 shows the numerical results of all comparison methods over all the four test network topologies.

*4.2.1 Effectiveness of Solution via Optimal Transport.* In the small-scale test network scenario, e.g., B4, *Tombo* takes only 2.69 seconds (5× speedup) on average to obtain a traffic allocation which is nearly the same as the best-performing scheme Tombo-Gurobi. *Tombo* also outperforms Tombo-Gurobi and MCF-Gurobi in networks Janetbackbone and Carnet in terms of the running time, with a speedup of 7× and 6×, respectively. In addition, *Tombo* has a remarkably competitive performance in terms of total objective values compared to Tombo-Gurobi in four test network topologies, with an average solution gap of less than 2.5%. In the UsCarrier network,

**Table 2: Performance comparison in terms of different evaluation metrics in four test network topologies.**

| Topology | Method | Objective | | | Violation ↓ | Running time (s) ↓ | Speedup↑ | Avg. bitrate (bpp) ↓ | Avg. link utilization↓ |
|---|---|---|---|---|---|---|---|---|---|
| | | Total Obj.↓ | Trans. cost↓ | Task-surr.↓ | | | | | |
| B4 | Tombo | 85.84 | **47.71** | 38.13 | 1.82e-12 | **2.69** | 4.98× | **0.1222** | **0.0502** |
| | Tombo-Gurobi | **85.73** | 49.03 | **36.70** | 7.14e-11 | 13.39 | 1× | 0.1255 | 0.0516 |
| | MCF-Gurobi (k=3) | 87.74 | 49.65 | 38.09 | 9.09e-13 | 55.11 | 0.24× | 0.1231 | 0.0523 |
| | TM-Sinkhorn | 89.77 | 49.74 | 40.02 | 0 | 3.10e-2 | 431.94× | 0.1222 | 0.0523 |
| | SRD-Sinkhorn | 89.27 | 45.49 | 43.79 | 4.55e-13 | 2.68 | 5.00× | 0.1170 | 0.0479 |
| Janetbackbone | Tombo | 282.24 | **122.48** | 159.76 | 7.63e-2 | **12.51** | 6.80× | **0.1349** | 0.0544 |
| | Tombo-Gurobi | **277.14** | 140.81 | **147.53** | 0 | 85.10 | 1 × | 0.1447 | 0.0575 |
| | MCF-Gurobi (k=3) | 277.47 | 128.44 | 149.03 | 1.25e-11 | 105.01 | 0.81× | 0.1436 | **0.0475** |
| | TM-Sinkhorn | 295.32 | 127.11 | 168.21 | 0 | 4.55e-2 | 1870.33× | 0.1349 | 0.0583 |
| | SRD-Sinkhorn | 305.86 | 136.14 | 169.72 | 9.70e-3 | 8.38 | 10.16× | 0.1331 | 0.0605 |
| Carnet | Tombo | 593.62 | 181.82 | 411.81 | 8.39e-2 | **94.83** | 5.80× | 0.1186 | 0.0701 |
| | Tombo-Gurobi | **557.86** | 160.43 | **397.42** | 9.09e-13 | 550.12 | 1× | 0.1318 | 0.0626 |
| | MCF-Gurobi (k=3) | 890.98 | 101.40 | 789.58 | 0 | 850.77 | 0.65× | **0.0917** | **0.0472** |
| | TM-Sinkhorn | 622.41 | 150.76 | 471.64 | 1.36e-12 | 0.91 | 604.53× | 0.1194 | 0.0704 |
| | SRD-Sinkhorn | 620.01 | 185.12 | 434.88 | 1.33e-2 | 43.06 | 12.76× | 0.1228 | 0.0719 |
| UsCarrier | Tombo | 311.85 | 169.39 | 142.45 | 1.36e-12 | **35.05** | 113.84× | **0.1265** | **0.0026** |
| | Tombo-Gurobi | **307.88** | 175.00 | **132.88** | 0 | 3990.00 | 1× | 0.1346 | **0.0026** |
| | MCF-Gurobi (k=1) | 322.14 | 151.34 | 170.80 | 0 | 1402.00 | 2.85× | 0.1179 | 0.0061 |
| | TM-Sinkhorn | 327.61 | 175.15 | 152.45 | 0 | 0.25 | 15960× | 0.1265 | 0.0049 |
| | SRD-Sinkhorn | 382.64 | 224.82 | 157.82 | 0 | 38.31 | 104.15× | 0.1246 | 0.0049 |

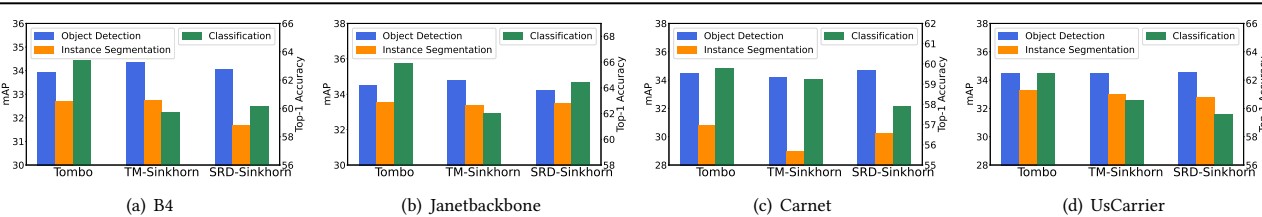

(a) B4     (b) Janetbackbone     (c) Carnet     (d) UsCarrier

**Figure 4: Comparison of task performance of Tombo, SRD-Sinkhorn, and TM-Sinkhorn across three different tasks.**

which has 158 nodes and 378 links, *Tombo* shows a significantly faster solving speed (114× speedup) compared to other comparison methods, while maintaining competitive performance, with the total objective gap less than 1.2%. The results indicate that the speedup performance compared to Tombo-Gurobi grows as the network scale increases.

Furthermore, we observe that all comparison methods achieve a solution with a tiny violation value, indicating that the link overload is negligible. Moreover, *Tombo* obtains comparable results for the average bitrate and link utilization as Tombo-Gurobi. Note that MCF-Gurobi requires a longer time compared to Tombo-based approaches in order to achieve the solution, even when the time used to obtain preset pathways is excluded. This demonstrates the efficacy of converting MCF problems into multi-marginal optimal transport problems as it greatly speeds up problem-solving processes. Furthermore, MCF-Gurobi simply distributes traffic along the shortest $k$ paths for each source-target node pair, without fully using all the available network links. Consequently, there is a decline in the performance of the total objective value.

*4.2.2 Effectiveness of Joint Bitrate and Routing Optimization.* We then investigate the impact on the overall task performance by introducing task-specific performance metrics and the joint optimization framework. Through comparison of the task performance achieved by *Tombo* with the baseline methods TM-Sinkhorn and SRD-Sinkhorn, it is demonstrated that *Tombo* achieves the superior performance in terms of the task performance over all networks. As shown in Fig. 4, TM-Sinkhorn and SRD-Sinkhorn present a significant degradation in the overall task performance because they do not distinguish between different downstream tasks when optimizing bitrates. On the four network topologies, *Tombo* consistently

**Table 3: Performance of Tombo-D vs. Tombo in dynamic network scenarios with $\rho_{vl} = 5\%$ and $q = 2.5$.**

| Topology | Objective | | Running time (s) | | Speedup |
|---|---|---|---|---|---|
| | Tombo-D | Tombo | Tombo-D | Tombo | |
| B4 | 86.17 | 86.17 | 1.75 ± 2.04 | 2.85 ± 1.55 | 1.62× |
| Janetbackbone | 309.05 | 309.05 | 10.04 ± 3.77 | 14.30 ± 1.31 | 1.42× |
| Carnet | 641.75 | 641.75 | 36.48 ± 28.69 | 82.51 ± 6.83 | 2.26× |
| UsCarrier | 311.85 | 311.85 | 35.83 ± 13.58 | 53.71 ± 0.93 | 1.50× |

demonstrates a fairer performance across different image tasks. Notably, for the other two methods, particularly TM-Sinkhorn in the Carnet network, the performance of the object detection task significantly surpasses that of instance segmentation. This observation can be attributed to the average distribution of bitstreams at the receiving end for different tasks. However, it is crucial to recognize that different image tasks impose distinct requirements on the image compression to achieve comparable performance levels.

*4.2.3 Adaptation to Dynamic Network Changes.* As shown in Table 3, the running time of *Tombo-D*, which uses the iterative variables from the last transmission period to initialize the current variables, is significantly shorter than that of *Tombo* across the four test networks scenarios. Overall, *Tombo-D* achieves a minimum 1.5× speedup in running time over *Tombo*. This finding highlights a strong capacity of *Tombo-D* to adapt to the time-varying network capacity. By initializing iterative variables using the intermediate variables from the solution of previous transmission period, *Tombo-D* speeds up the convergence while preserving the same level of accuracy as *Tombo*. These results also demonstrate that slight modifications in the capacity of network links have only a minimal effect on the feasible domain of the optimization problem. By re-using

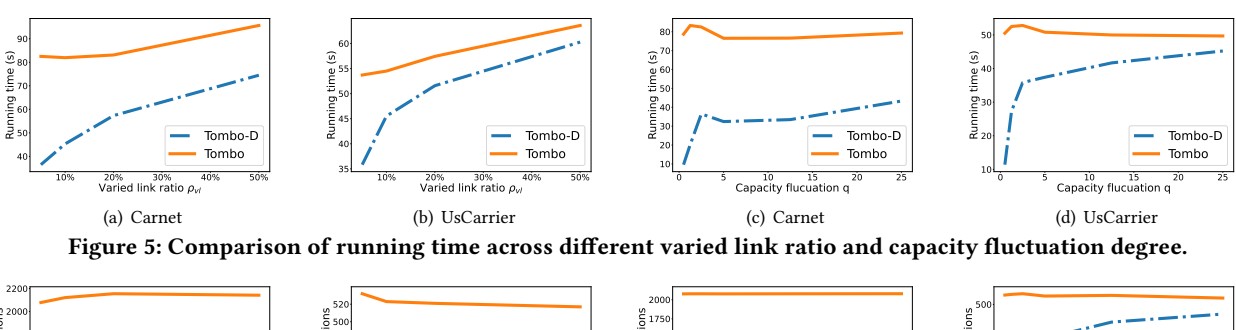

(a) Carnet     (b) UsCarrier     (c) Carnet     (d) UsCarrier

Figure 5: Comparison of running time across different varied link ratio and capacity fluctuation degree.

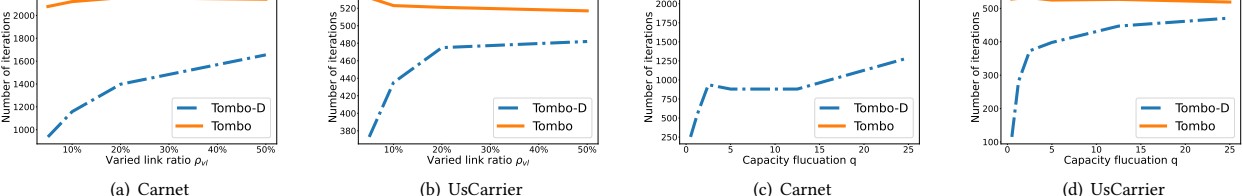

(a) Carnet     (b) UsCarrier     (c) Carnet     (d) UsCarrier

Figure 6: Comparison of iteration number across different varied link ratio and capacity fluctuation degree.

iterative variables from the previous solution, we provide an initial starting point for new iterations that is close to optimum solution.

## 4.3 Ablation Study

*4.3.1 Regularization Parameter.* In order to examine the influence of the regularization entropy term $\epsilon$ on the solution's quality, we compared *Tombo* using various values of $\epsilon$ in terms of the overall objective value, capacity violation, and running time. As depicted in Table 4, *Tombo* with the smallest $\epsilon$ consistently yields the closest approximation to the optimal solution. However, achieving a feasible solution with a smaller $\epsilon$ requires more time due to insufficient iterations, resulting in a significant violation of capacity constraints. Nevertheless, with the increase in the number of iterations, the violation of link capacity constraints gradually diminishes. Moreover, as $\epsilon$ increases, the time required for each iteration decreases. With a larger $\epsilon$, the objective value remains the same as the number of iterations increases, indicating that a higher $\epsilon$ fosters a swifter convergence of *Tombo*. This phenomenon can be attributed to the smoother boundaries of feasible regions. While *Tombo* with a large $\epsilon$ can swiftly compute a feasible solution, it deviates notably from the optimum. Notably, in comparison to B4, which comprises only 12 nodes, the disparity between the solution and the optimum is more pronounced in larger-scale networks like Janetbackbone.

*4.3.2 Link Capacity Fluctuation.* To assess the impact of network link capacity fluctuations on the performance of *Tombo-D*, we conduct experiments focusing on two key factors: the proportion $\rho_{vl}$ of selected links where the capacity changes, and the corresponding threshold of link capacity fluctuation $q$. We first conduct the experiments with the settings $\rho_{vl} = \{5\%, 10\%, 20\%, 50\%\}$ with fixed $q = 2.5$, and then adjusted values of $q = \{1.25, 2.5, 5, 12.5, 25\}$ with fixed $\rho_{vl} = 5\%$. The comparison result between *Tombo-D* and *Tombo* in terms of the running time and iteration number in the Carnet and UsCarrier networks are illustrated in Figs. 5 and 6, respectively. It is seen that *Tombo-D* can expedite iterative convergence and save computational time across different proportions of varying links in the two test scenarios. This phenomenon can be attributed to the effective initialization of the iteration variables, facilitating faster convergence and the discovery of optimal solutions. Nonetheless,

Table 4: Ablation results w.r.t. regularization parameter $\epsilon$.

| Topology | $\epsilon$ | Iteration | Total Obj. | Violation | Running time (s) |
|---|---|---|---|---|---|
| B4 | 5e-4 | 50 | 86.16 | 1.73 | 0.2488 |
| | | 200 | 86.17 | 1.15e-8 | 1.0455 |
| | | 500 | 86.17 | 2.27e-13 | 2.1639 |
| | 1e-3 | 50 | 87.62 | 4.30e-3 | 0.1767 |
| | | 200 | 87.62 | 2.27e-13 | 0.6165 |
| | | 500 | 87.63 | 2.27e-13 | 1.4488 |
| | 5e-3 | 50 | 105.53 | 0 | 0.1089 |
| | | 200 | 105.53 | 0 | 0.3321 |
| | | 500 | 105.53 | 0 | 0.8360 |
| Janetbackbone | 5e-4 | 50 | 279.61 | 560.51 | 0.8917 |
| | | 200 | 281.98 | 33.61 | 3.7333 |
| | | 500 | 282.23 | 1.19 | 10.1919 |
| | 1e-3 | 50 | 297.96 | 81.31 | 0.6150 |
| | | 200 | 298.48 | 2.01 | 2.4022 |
| | | 500 | 298.48 | 2.90e-3 | 6.2031 |
| | 5e-3 | 50 | 368.14 | 2.21e-7 | 0.4025 |
| | | 200 | 368.14 | 0 | 1.5481 |
| | | 500 | 368.14 | 0 | 3.6269 |

as the value of $\rho_{vl}$ rises, the advantage in running time diminishes. This phenomenon is reasonable since in the extreme scenario where all link capacities change, a completely new optimization problem emerges. As shown in Figs. 5(c) and 5(d), *Tombo-D* demonstrates a notable acceleration across all levels of link capacity fluctuations. Nevertheless, as network link capacities undergo more pronounced changes, the running time advantage diminishes. This phenomenon arises because the starting point of iterations in the feasible region becomes further away from the updated optimal solution.

## 5 CONCLUSION

We have proposed a framework for jointly optimizing the encoding bitrates and routing scheme for multiple image bitstreams with various machine vision tasks, named *Tombo*. The task-oriented multi-bitstream compression and transmission problem was formulated as an MCF problem with time-expanded network modeling. To quickly solve the problem, we re-formulated the MCF problem to a multi-marginal optimal transport problem and proposed a Sinkhorn iteration-based algorithm to speed up the solving process. Then, we proposed the enhanced *Tombo-D* to adapt to dynamic networks where link capacities fluctuate over time. Evaluations on three typical machine vision tasks and four real-world network topologies have demonstrated the effectiveness of *Tombo* and *Tombo-D*.

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
