# OpenReview forum: "Task-Oriented Multi-Bitstream Optimization for Image Compression and Transmission via Optimal Transport"
_acmmm.org/ACMMM/2024/Conference — MM2024 Poster_

### Official Review · Reviewer_guVi · 2024-05-13

**Rating:** 2
**Confidence:** 4

**Summary:**

This paper proposes a framework for jointly optimizing the encoding bitrates and routing scheme for multiple image bitstreams with various machine vision tasks. They formulated the problem as a multi-marginal optimal transport problem and proposed a Sinkhorn iteration-based algorithm to speed up the solving process.

**Strengths:**

1.	This paper has a good theoretical approach.
2.	The figures in this paper are of good quality and easy to understand.

**Limitations:**

1.	There have been many research works on image compression and transmission methods. This paper Lacks a presentation of related work and no benefits from it. I suggest increasing a new discussion to show what are the limitations of the related works.
2.	The authors mainly compared with some optimization algorithm solvers and lacked a comparison of image compression and transmission schemes. The author should compare the proposed algorithm with other recent works or provide a discussion. Otherwise, it's hard for the reader to identify the novelty and contribution of this work.
3.	This paper focuses on the joint optimization problem of image compression and transmission but lacks a detailed description of the image compression scheme.
4.	Some of the methods and experiments presented in this paper are relatively independent and lack end-to-end performance comparisons. Time is a very important performance metric in the study of image compression and transmission. It is recommended that the authors add a compression and transmission time comparison with baseline methods.
5.	This paper is mainly oriented towards different tasks, including object detection, classification, and instance segmentation. However, it does not present the specific methods or models used.
6.	The results in Table 2 are very confusing. The criteria for bolding are inconsistent, for example, in “Trans. cost” why 47.71 is bolded instead of 45.49. The authors are advised to double-check.
7.	The results in Table 2 show the Speedup of TM-Sinkhorn is 604.53, but the method of this paper is only 5.80. It is suggested that the authors give a reasonable explanation, otherwise it is difficult to judge the contribution of this paper.
8.	There are some minor errors, such as “Fig. 1” should be “Figure 1”, and “COCO2017” should be “COCO 2017”.
9.	The references are not in a consistent format, such as [16], [20], [25].

**Suitability:**

2

---

### Official Review · Reviewer_VcR4 · 2024-05-25

**Rating:** 4
**Confidence:** 3

**Summary:**

This well-written paper, featuring rigorous mathematical derivations, addresses the varying rate-accuracy performance of image compression for different machine vision tasks and content types. In this paper, the authors propose Tombo, a task-oriented image compression and transmission framework that identifies the optimal encoding bitrate and routing scheme for multiple image bitstreams delivered simultaneously for different downstream tasks. They study the characteristics of image rate-accuracy performance for different machine vision tasks and formulate the task-oriented joint bitrate and routing optimization problem for multi-bitstreams as a multi-commodity network flow problem with time-expanded network modeling. To ensure consistency between encoding bitrate and routing optimization, the authors propose an augmented network that incorporates encoding bitrate variables into the routing variables. To improve computational efficiency, they convert the original optimization problem to a multi-marginal optimal transport problem and adopt an existing Sinkhorn iteration-based algorithm to obtain solutions quickly. They also extended Tombo to handle dynamic network scenarios where link capacities fluctuate efficiently. Simulations with three machine vision tasks and four real-world network topologies demonstrate that Tombo achieves comparable performance to the optimal solution provided by a solver, Gurobi, with a 5× to 114× speedup.

**Strengths:**

+ Well-written and very easy to follow.

+ Detailed math derivations, although the major results are based on prior arts, such as:
Gabriel Peyre and Marco Cuturi. 2019. Computational Optimal Transport. Foundations and Trends in Machine Learning 11, 5-6 (2019), 355–607.

+ Detailed simulation results.

**Limitations:**

- I missed the related work section. The problem is not entirely new; for example, it has been thoroughly studied in visual-based wireless networks. How would those works differ from the current paper? The authors are suggested to cite at least a few survey paper, like:

B. A. Lungisani, C. K. Lebekwe, A. M. Zungeru and A. Yahya, "Image Compression Techniques in Wireless Sensor Networks: A Survey and Comparison," in IEEE Access, vol. 10, pp. 82511-82530, 2022, doi: 10.1109/ACCESS.2022.3195891.
keywords: {Image coding;Wireless sensor networks;Data compression;Wireless communication;Sensors;Discrete cosine transforms;Principal component analysis;Autoencoder;data compression;image compression;image compression techniques;lossy compression;wireless sensor network(s)},

Along the same line, all the baselines are only variants of the proposed solution. It will be better if some prior algorithms can be compared against.

- Is there a theoretical bound for the near-optimality?

- How practical is the proposed solution? (1) where and how often is the algorithm executed? (2) How to enforce the routing decisions? (3) What are the typical workloads (request arrival patterns)? (4) How is background traffic considered? (5) Do all nodes in the considered problem need to be synchronized?

- Evaluations only consider very few sources/targets. Does the proposed algorithms scale to bigger networks?

- "Fig. 2(a) shows a common scenario where a sequence of compressed image signals are requested to be transmitted from a source
(e.g., a monitor) to several targets to..." a monitor? Not a camera?

- Sec. 2.2: all tasks are assumed to be equally important?

- Sec. 2.2: How the Φ𝑙,𝜃 (𝑟𝑙 ) = −𝑎𝑙,𝜃 /(𝑟𝑙 − 𝑏𝑙,𝜃 ) + 𝑐𝑙,𝜃 chosen? Are there other candidates?

- Sec. 4: How were network protocol and other overheads considered?

**Suitability:**

2

---

### Official Review · Reviewer_ARBd · 2024-05-25

**Rating:** 3
**Confidence:** 3

**Summary:**

The paper propose a method to find the optimal decision on encoding bitrate and routing for machine vision transmission tasks. In conclusion, the paper really focus on a interesting problem, but it does not provide a good and convincing solution for the problem. The innovation of this paper is to introduce multipath routing into the scenario of machine vision task transmission, but the author lacks sufficient research on state-of-the-art related work about multipath routing. What's more, the author needs to explain the challenges and difficulties of this work before starting introducing the methodology. Last, the author considers this problem from a theoretical perspective, which may be difficult to apply in practical scenarios because of the difficulty of hypothesis on fluctuating network information. It is strongly recommended that the author consider a better solution from the perspective of the actual system.

**Strengths:**

- The paper focuses on a very interesting issue, which has good research value for both the academic community and practical applications.

- The paper is well-written and the problem is clearly stated.

**Limitations:**

main concerns:
- lack of survey on related work. The only work discussed is a paper from ICNC 2018, which is very insufficient. However, there are currently a large number of traditional multipath routing works (such as MPTCP, MPQUIC, MPRTP), and in recent years, many works have also discussed how to achieve higher bitrate  and QoE (Raven Mobicom'18, XLink in sigcomm 21, Converge in sigcomm 23 and so on).
- the experiment baseline is too simple, just compare with optimization solver, lack comparison of the state-of-the-art routing framework

minor problems:
- "Though off-theshelf solvers, e.g., Gurobi [15] and CPLEX [10], can accurately solve this problem to get the optimal solution, they cannot scale to the growing number of network nodes and commodities" why do you need consider growing number of network nodes? In fact most multipath routing just have 2-3 available paths to use.
- I was wonder whether you can choose path between the same sender and receiver, i.e. xp11, xp33 in the example shown in figure 2, unless you are talking about sending with different NICs. Since the routing decision between the same sender and receiver is handled by ISP and Internet router, the only thing sender can do is build path on different NICs. But in the case of machine vision task, what device are using? Do they really support multiple NICs?
- there are too many notations, the authors should add a table in the text or appendix. And there are still some confusing expressions need to clarify:
    - The relation between bitstream l and task \theta. Are they the same? If they are different, why?
    - If the total bitrate of edge should be less than total bandwidth (Eqn.2d), why need the variable r? The variables of xp and r are redundent.
- Also, there are some doubts about this optimization function：If there are many sources to send images, and each source has to send multiple images, which is pretty common in practice, how to deal with the asynchronous sending problem? After all, the service can not wait for all the sources to finish its sending and then allow the next round sending.

**Suitability:**

3

---

### Official Review · Reviewer_1vuX · 2024-05-27

**Rating:** 5
**Confidence:** 3

**Summary:**

The paper proposed a task-oriented image compression and transmission framework called Tombo, which efficiently identifies the optimal encoding bitrate and routing scheme for multiple image bitstreams delivered simultaneously for different downstream tasks.

**Strengths:**

One strength of this paper is the comprehensive approach taken to address the task-oriented multi-bitstream optimization problem in the context of image compression and transmission for machine vision tasks. The paper proposes the Tombo framework, which efficiently identifies the optimal encoding bitrate and routing scheme for multiple image bitstreams delivered simultaneously for different downstream tasks.
Specifically, the paper formulates the optimization problem as a multi-commodity flow problem with time-expanded network modeling, taking into account the characteristics of image rate-accuracy performance for different machine vision tasks. The use of a multi-marginal optimal transport approach and the development of a Sinkhorn iteration-based algorithm contribute to the quick and near-optimal solution finding process.
Furthermore, the paper addresses the dynamic network scenario where link capacities may fluctuate over time by enhancing Tombo to adapt rapidly to these changes. The proposed Tombo-D framework leverages solutions obtained in the previous transmission period to improve computational efficiency in dynamic network scenarios.
Overall, the paper provides a detailed and innovative solution to the task-oriented image compression and transmission problem, demonstrating effectiveness in terms of overall task performance and computational efficiency through empirical evaluations on real-world network topologies and machine vision tasks.

**Limitations:**

One limitation of this paper could be the focus on empirical evaluations on a limited set of machine vision tasks and network topologies. While the proposed Tombo framework shows effectiveness in these specific scenarios, the generalizability of the approach to a wider range of tasks and network configurations may not have been fully explored.

This paper mentions image compression in both the title and the introduction part of the main text, but in fact, the core content of the paper is the intelligent transmission of multi-task streams, which has little to do with image compression. In addition, in the experimental part, it is not clearly written what image compression and visual task network are used, and the experimental results about the bitrate and accuracy of image compression are also very few, which is not consistent with the overall joint optimization framework mentioned in the title. It is suggested to modify the description in the title and body part to weaken the image compression and focus on the transmission part.

**Suitability:**

2

---

### Meta-Review · Area_Chair_mtbi · 2024-07-03

**Recommendation:** Accept (Poster)
**Confidence:** 3

**Metareview:**

This paper presents a task-oriented image compression and transmission framework  that exposes encoding information at the routing layer. All the reviewers appreciate the technical contribution of the paper and find its contribution valuable. The evaluations range from weak reject to weak accept. Reviewers find the related work and the proper context for the paper to be somewhat missing. I recommend acceptance based upon the solid theoretical foundations and valuable proposal of this paper.